# Safety and Efficacy of a Single-Stage versus Two-Stage Intramedullary Nailing for Synchronous Impending or Pathologic Fractures of Bilateral Femur for Oncologic Indications: A Systematic Review

**DOI:** 10.3390/cancers15174396

**Published:** 2023-09-02

**Authors:** Patrick P. Nian, Vanathi Ganesan, Joydeep Baidya, Ryan S. Marder, Krish Maheshwari, Andriy Kobryn, Aditya V. Maheshwari

**Affiliations:** Department of Orthopaedic Surgery and Rehabilitation Medicine, State University of New York Downstate Health Sciences University, Brooklyn, NY 11203, USA

**Keywords:** intramedullary nailing, bilateral femur, impending fracture, pathologic fracture

## Abstract

**Simple Summary:**

For patients with advanced cancer presenting with synchronous bilateral femur impending and/or complete pathologic fractures requiring placement of intramedullary nails (IMN), the optimal timing of bone fixation—whether in single-stage (SS) or two-stage (TS)—is still highly debatable. In this study, we systemically reviewed and compared the existing literature for perioperative outcomes such as complications, survival, same-admission mortality, length of stay, and start of rehabilitation and adjuvants between the SS and TS groups. Contrary to old literature, our findings revealed that SS IMN in select patients does not increase their risk of perioperative complications and same-admission mortality. However, limited comparative data exist on other proposed benefits of SS IMN including length of stay, earlier start of rehabilitation and adjuvant therapy, functional scores, and cost. Our results support SS bilateral femur IMN as a safe and efficient strategy for select patients.

**Abstract:**

Although intramedullary nail (IMN) fixation is the standard of care for most impending and/or complete pathologic fractures of the femur, the optimal timing/sequence of the IMN in cases of synchronous bilateral femoral disease in advanced cancer is not well established. Thus, we compared the outcomes of single-stage (SS) vs. two-stage (TS) IMN of the bilateral femur with a systematic review of the literature on this topic. Bilateral SS and TS IMN cases were identified from 14 studies extracted from four databases according to PRISMA guidelines. Safety (complications, reoperations, mortality, survival, blood loss, and transfusion) and efficacy (length of stay [LOS], time to start rehabilitation and adjuvant therapy, functional scores, and cost) were compared between the groups. A total of 156 IMNs in 78 patients (36 SS and 42 TS) were analyzed. There were one surgical (infection in TS requiring reoperation; *p* = 0.860) and fifteen medical complications (five in SS, ten in TS; *p* = 0.045), with SS being associated with lower rates of total and medical complications. Survival, intraoperative mortality, and postoperative same-admission mortality were similar. No cases of implant failure were reported. Data on LOS, rehabilitation, and adjuvant therapy were scarcely reported, although one study favored SS over TS. No study compared cost or functional scores. Our study is the largest and most comprehensive of its kind in supporting the safety and efficacy of a SS bilateral femur IMN approach in these select patients. Further investigations with higher levels of evidence are warranted to optimize treatment protocols for this clinical scenario.

## 1. Introduction

Modern advances in cancer diagnosis and treatment have increased survival rates, which has subsequently resulted in a higher incidence and prevalence of metastatic disease to the bone, including synchronous involvement of multiple long bones [1,2,3,4,5]. The femur is the most common long bone involved and intramedullary nail (IMN) fixation is widely considered the surgical standard of care for most patients as it reduces hospital length of stay, promotes patient’s independence at the end of life, aims to relieve pain, maintains and restores early postoperative weight-bearing, mobility, and function, expedites adjuvant treatment of the primary malignancy, and improves patients’ quality of life [2,6,7,8,9,10,11,12,13,14,15,16].

However, IMNs of even unilateral femurs pose significant risks, including fat and tumor emboli, infection, thromboembolism, hardware failure, and cardiopulmonary complications leading to death [12,13,17,18,19,20,21,22,23,24,25,26]. Thus, when patients present with synchronous bilateral femoral disease, the timing of two IMNs, single-stage (SS) vs. two-stage (TS), becomes an important decision to balance the minimization of these complications but to also expedite the definitive oncologic treatment. Due to the increased theoretical thromboembolic and cardiopulmonary events with bilateral IMN, historically TS was the preferred approach for bilateral femur IMN [25,26,27,28]. However, some recent studies have shown encouraging results with SS bilateral IMN citing potential advantages of a single anesthesia, early start of rehabilitation and adjuvants, reduced length of stay, and most importantly, non-inferior clinical outcomes [1,3,7,29,30,31].

Despite being such an important clinical and decision-making question, there is no study that directly compares the SS vs. TS approach for bilateral femur IMN, and only one systemic review [7] has investigated survival time and perioperative complications in only 17 patients. Therefore, any meaningful conclusion has been limited by the existing study designs including relatively small sample sizes, heterogeneity, and lack of comparative groups, and thus the question on the optimal surgical timing, SS vs. TS, is still debatable.

The objective of our study was to conduct an updated systematic review of existing literature to analyze and compare the safety and efficacy of a SS approach to a TS approach for patients presenting with synchronous bilateral impending and/or pathologic fracture of the femur for oncologic indications. Considering recent advancements in the surgical care and perioperative management of these patients, we hypothesized that SS will not be inferior to the TS approach with respect to the aforementioned outcome measures but may have some clinical advantages.

## 2. Materials and Methods

### 2.1. Literature Search

Medline/PubMed, EMBASE, Scopus, and the Web of Science databases were systematically searched from the inception of these databases until February 2023. The Boolean search terms and operators used for each search are as follows: [nail* AND (intramedullary OR IM OR IMN OR cephalomedullary) AND (bilateral OR stage* OR simultaneous OR single) AND femur* AND (orthopedic* OR orthopaedic*)]. This exact query was entered as is when conducting the literature search on each of the databases utilized. The usage of four independent databases and multiple permutations of search terms minimized any intentional risk of bias when selecting manuscripts to analyze. The results of our systematic review are reported according to the Preferred Reporting Items for Systematic Reviews and Meta-Analysis (PRISMA) 2020 statement. Our systematic review is not registered on PROSPERO.

### 2.2. Eligibility Criteria

Studies were eligible for inclusion if they met the following criteria: (1) full-text publications reporting on the safety, efficacy, and outcomes of bilateral (SS or TS) femoral IMN procedures for metastatic bone disease or multiple myeloma in adult patients; and (2) articles published in English. Studies were excluded if IMNs were performed for non-oncologic indications, metachronous bilateral femur disease and/or performed during different admissions (defined as >12 weeks if not specified), data were insufficient or unavailable, non-IMN fixation methods (e.g., arthroplasty, plating, dynamic hip screw, isolated cementoplasty, flexible nails, or tumor prosthesis) were utilized, and if details regarding staging and outcomes were unspecified. All review articles, expert opinions, editorials, commentaries, biomechanical studies, book chapters, epidemiological/incidence reports, and technical notes were excluded.

### 2.3. Study Selection

After duplicate article exclusion, titles and abstracts were independently screened by two investigators (RSM and AK) according to the inclusion criteria. Any discrepancies were resolved by the senior investigator (AVM). Full-text review was performed on all studies after the screening process and on any additional studies where uncertainty was encountered to further determine inclusion eligibility. The data-relevant outcomes of bilateral femoral nailing procedures for oncologic indications were independently extracted from the included publications. The information (if reported) retrieved from each publication included the following: (1) general study information (e.g., author, title, study design, year of publication); (2) SS vs. TS; (3) patient demographics (e.g., age, gender, primary tumor, etc.); (4) details of the surgical procedure; and (5) outcome measures of patient safety and efficacy.

### 2.4. Surgical Procedure Data

The surgical procedure measures extracted from each publication, when available, included the following: (1) type and size of nail used; (2) surgical technique (e.g., canal venting, diaphyseal reaming, cement use, distal locking screw utilization, reaming irrigation aspiration [RIA]; (3) type of anticoagulation used; (4) time delay between the first and second femoral IMN for TS cases; and (5) cases of aborted SS or deaths in between planned TS bilateral femoral IMN.

### 2.5. Patient Safety Data

The measures of patient safety extracted from each publication, when available, included the following: (1) perioperative medical complications, including cardiopulmonary complications with or without embolic events, and other systems-based complications as reported; (2) surgical complications including implant-related complications and reoperations and others as reported; (3) same-admission (including intraoperative and postoperative) mortality; (4) survival; and (5) blood loss and blood transfusion (up to 24 h postoperatively).

### 2.6. Surgical Efficacy Data

The measures of efficacy extracted from each publication, when available, included (1) length of stay (LOS); (2) start of rehabilitation and definitive oncologic adjuvant therapy; (3) functional scores; and (4) cost.

### 2.7. Statistical Analysis

When available, statistical analyses were conducted, including chi-squared and Student’s *t*-tests to compare categorical and continuous variables, respectively, and Kaplan–Meier estimates with the log-rank test for patient survivorship. All analyses were performed in R Statistical Software (R version 4.3.0, Foundation for Computational Statistics; Vienna, Austria) using a *p*-value of < 0.05 as the threshold for statistical significance.

## 3. Results

### 3.1. Study Selection and General Characteristics

The initial online literature search resulted in 1972 publications (418, 1028, 252, 261, and 13 in Medline/PubMed, EMBASE, Scopus, Web of Science, and other sources, respectively). After the screening process was completed, 70 articles were selected for full-text review and analysis. Of those, 14 studies [1,3,17,26,29,30,31,32,33,34,35,36,37,38] were included in our systematic review. The Preferred Reporting Items for Systematic Reviews and Meta-Analyses (PRISMA) flowchart is depicted in Figure 1.

The studies included in our systematic review reported on 156 intramedullary nails placed in 78 patients. The average age of patients undergoing bilateral procedures was 62.3 (32–82) among the reported studies. Out of 78 patients, 36 (46.2%) underwent SS bilateral femoral nailing and 42 (53.8%) underwent TS bilateral femoral nailing procedures, but no specific objective selection criteria were described, except for one study that made these decisions with a multidisciplinary team, taking into consideration the patient’s tumor burden, general medical condition, rehabilitation potential, timing of adjuvant therapeutic modalities, and patient’s and/or their family’s wishes [31]. General characteristics of the included studies are summarized in Table 1. Seven studies [17,26,29,31,32,34,37] reported the average time delay between the first and second femoral IMN for bilateral procedures, excluding those staged more than 12 weeks apart, as 10.8 ± 9.9 [range, 3–56] days (Table 2). Three studies noted the type of anticoagulation [17,31,33] as aspirin, coumadin, heparin, low-molecular-weight heparin, or inferior vena cava filter. There were no reported aborted cases in SS in the literature and no deaths in between planned procedures in TS. 

### 3.2. Surgical Technique and Implant Considerations

Of the fourteen included studies, eight reported data on the type of nail, eleven studies reported on the type of reaming used, and four studies reported on nail size. No study showed any difference in outcomes with nail type (solid vs. hollow), IMN technique, reaming, or venting (Table 2).

For both SS and TS unreamed nailing cases, there were no reported cardiopulmonary complications or intraoperative deaths [30,36,37]. Perioperative cardiovascular complications (including thromboembolism, cardiac arrests, etc.) were seen in at least three cases intraoperatively of cardiac arrest in the SS reamed group, where there was one intraoperative death and two successful resuscitations, and one case of postoperative respiratory distress from presumed fat bolus [3,26,29,33,35,38]. In the TS reamed group, there were two cardiac arrests due to air and fat emboli, both leading to intraoperative death [30,36,37] (Table 2).

### 3.3. Patient Safety

#### 3.3.1. Medical and Surgical Complications

There were significantly more total complications reported in the TS as compared to the SS cohort, which were mostly medical. Nine publications reported measures of patient safety (medical and surgical complications) for 34 and 24 patients in the SS and the TS group, respectively [1,3,26,30,31,33,35,37,38] (Table 3). Total complications were reported in five of thirty-four (14.7%) in the SS and eleven of twenty-four (45.8%) patients in the TS group (*p* = 0.021) and were mostly medical complications [5 (14.7%) vs. 10 (41.6%); *p* = 0.045].

Cardiopulmonary complications were the most commonly reported medical complications in each group, with no statistically significant differences in the proportion of cardiopulmonary to medical complications between the SS and the TS cohort [5/5 (100%) vs. 7/10 (70.0%); *p* = 0.494]. Further, of the five medical complications in the SS group, two (40%) led to same-admission mortality (one intraoperative and one postoperative) while in the TS group, two (20%) led to same-admission mortality (two intraoperative and zero postoperative).

#### 3.3.2. Same-Admission Mortality and Survivorship

Length of survival was reported in seven studies for 25 and 19 patients in the SS and the TS group, respectively, and did not differ significantly between the two cohorts when reported for 30 days or longer (*p* = 0.530) or 90 days or longer (*p* = 1.000) [1,3,26,29,31,34,35] (Table 3). We also found no difference in the Kaplan–Meier survivorship between the two cohorts (*p* = 0.95) (Figure 2).

There was no significant difference in rates of intraoperative (*p* = 0.893) [1,26,35], postoperative (*p* = 0.778) [3,31], and total same-admission (intraoperative + postoperative) (*p* = 0.513) [1,3,26,31,35] mortality between the SS and the TS cohort (Table 3).

#### 3.3.3. Blood Loss and Transfusion

There was no significant difference in blood loss between the SS and the TS cohort. Data on blood loss were reported in two studies for eight and twenty-one patients in the SS and the TS cohort, respectively, and showed no significant difference between the cohorts (*p* = 0.211) [31,38] (Table 3). Other studies reported blood loss within their whole cohort, but these were not specific to patients undergoing bilateral femoral IMN.

No definitive comparison could be made regarding blood transfusion between SS vs. TS cohorts, as data were not clearly available in any study, or specified for bilateral femurs.

### 3.4. Efficacy

#### 3.4.1. Length of Stay

LOS has been shown to be shorter in the SS cohort compared to the TS cohort in one study. Data on LOS were reported in two studies [1,31], but only one [31] compared SS vs. TS and showed shorter stays for the SS group (7.3 ± 4.5 [range, 1–14] days) vs. TS (21.3 ± 18.1 [range, 3–65] days) [*p* = 0.006].

#### 3.4.2. Rehabilitation and Adjuvant Therapy

No definitive comparison could be made regarding time to rehabilitation between SS vs. TS cohorts as time to start was not specified for femurs.

Time to adjuvant therapy between SS vs. TS cohorts was difficult to determine because it was not consistently mentioned across the analyzed studies. However, one study commented that several patients had their definitive oncologic care at different institutions, which may have influenced the return to medical therapy rather than the surgical procedure itself [31].

#### 3.4.3. Functional Scores and Cost

Functional scores or cost were not able to be compared between the two cohorts as no papers reported on these data.

## 4. Discussion

Patients with cancer diagnoses are living longer than before due to advances in early diagnosis and treatment, and this has resulted in an increased incidence and prevalence of metastatic bone disease [2,4,34,39]. As the femur is the most commonly involved bone in the appendicular skeleton, it has become more common for patients to present with synchronous pathologic or impending bilateral femur fractures [5,39]. Although IMN is widely considered the preferred treatment modality, there still remains controversy concerning the optimal timing regarding fixation of the bilateral femora, SS vs. TS, in spite of some recent small case series reporting encouraging results with a SS approach [1,3,7,26,29,30,31]. Nevertheless, convincing evidence for the preferred staging approach remains lacking, as most existing studies are limited by small sample sizes, heterogeneity, and non-comparative designs. Therefore, the objective of this study was to conduct a systematic review of studies that investigated measures of patient safety and efficacy of SS or TS IMN for patients with synchronous impending or pathologic fractures of bilateral femora secondary to oncologic indications.

### 4.1. Surgical Technique and Implant Considerations

The most commonly used nails across the studies were the long gamma nail (LGN) and the solid AO nail. While the LGN was used across the studies for ten patients and the AO nail for nine patients with SS and TS surgical approaches, there is a paucity of data and comparisons between different types of intramedullary nail devices to determine whether one specific nail is associated with superior patient safety and functional outcomes.

The impact of diaphyseal reaming on complications and mortality in both SS and TS femoral IMN is not clear in our analysis nor in a previous study [3,7]. We focused on diaphyseal reaming as almost all nails need proximal reamings to accommodate their larger proximal diameter. Diaphyseal reaming in the tight isthmus may theoretically be associated with increased pressures locally within the femoral canal which has been linked to complications such as air, tumor, and fat emboli resulting in high mortality rates in some oncologic literature [26,29,40]. Thus, some have advocated the use of unreamed femoral nails for oncologic indications [30], although comparable mortality between reamed and unreamed IMNs has been shown in both oncologic and trauma literature [29,41,42].

Although distal femoral canal venting has been suggested as a surgical technique to reduce medullary canal pressures during diaphyseal reaming [1,29,30,43,44], it did not show any association with complications in the reported studies. As the fracture and/or open curettage site can act as a natural vent, this may be more important for prophylactic fixation. Kerr et al. [26] reported to have a 5 mm distal vent used during a TS prophylactic surgery, but this patient died intraoperatively due to a fat embolus and subsequent cardiac arrest. Moon et al. [3] described venting in two patients with multiple SS prophylactic long bone IMNs (out of 1 femur–femur and 4 femur–humerus) but did not mention specifically for bilateral femur cases, nor specified complications, if any. Although in vitro studies [43] show significantly decreased intramedullary pressures with venting, in vivo studies demonstrate this decrease from venting is not high enough to prevent emboli formation [3,44,45]. Based on these data, the clinical role of canal venting remains unclear, and many surgeons do not practice venting as a routine [1,3,31]. Although newer techniques like RIA [31,46,47] can theoretically reduce the risks of reaming, there is no study that has compared its use in SS vs. TS for oncologic indications.

Similarly, since no cases of hardware failure were reported in this study, no recommendation can be made about one vs. two distal locking screws, spiral blade vs. lag screw, or the size and type of nail, as again, the decision depends on the surgeons’ experience, available resources, and tumor location and burden [30,31,33,34].

The time between the two femoral IMNs in the TS group was reported as 10.79 ± 9.95 [range, 3–56] days. Kerr et al. [26] reported two patients with staged intervals less than 2 weeks. Both patients died intraoperatively during the second operation, and it was recommended that the second nailing should be delayed as long as possible [26]. Charnley et al. [25] similarly suggested a minimum of 2 weeks before the second femoral nailing should occur. In contrast, others [29,31] demonstrated no significant perioperative or immediate postoperative complications for those who underwent the second femoral nailing procedure within 2 weeks of the first. Since an earlier surgery may help with the early start of rehabilitation and definitive oncologic adjuvants [31], a balance is needed. However, determination of optimal timing between the two surgeries is not clear as it depends on multiple factors and is difficult to decipher from existing studies due to limited, unclear, and heterogeneous data. Moreover, the selection of patients for SS vs. TS seems biased [22,31,48] and may influence the timing. Thus, it should be tailored to each patient’s unique clinical situation and needs.

### 4.2. Patient Safety

#### 4.2.1. Medical and Surgical Complications

We found that the rate of total complications (mostly medical) was significantly higher in the TS compared to the SS cohort. However, it should be noted that there was a high degree of variability in the classification and severity of the complications reported in the included studies [12,17,18]. As expected, most of the medical complications in the study were cardiopulmonary in nature, and only one study [31] reported non-cardiopulmonary complications, which may suggest under-reporting of other complications deemed as less severe and/or concerning. Furthermore, there is likely a selection bias as patients selected for TS may have at baseline, a greater tumor burden and more systemic comorbidities. Future studies would benefit from a more uniform reporting or definition of complications. Notably, in our review of 34 patients in the SS cohort for whom complications data were available, we found three intraoperative cardiac arrests associated with fat/tumor/pulmonary emboli, one of which led to intraoperative death and two who were successfully resuscitated (Table 3). In the 24 patients in the TS cohort, we found two intraoperative cardiac arrests secondary to air/fat emboli, both of which led to intraoperative death. Other cardiopulmonary complications included one case of intraoperative hypotension requiring resuscitation and one case of postoperative respiratory distress secondary to presumed fat emboli, which led to death on hospital day 15 in the SS cohort and one case of intraoperative hypotension requiring resuscitation and four postoperative cases of pleural effusion, pneumonia, respiratory distress, and pulmonary embolism in the TS cohort [49].

No significant differences in surgical complications between the SS and the TS cohort were found. There was only one surgical complication reported in the TS group. This case of deep wound infection necessitated the only reported return to the operating room in our review, leading to surgical debridement after the second of a TS bilateral femoral IMN, as reported by Shemesh et al. [34]. No cases of implant failure, or resulting reoperation, were reported in our systematic review. Literature notes high loss to follow-up and only two papers indicated mean follow-up time of less than 6 months [32,34]. Failure of IMN for long bone metastases is typically an event with a median time of onset and revision that surpasses median survival in these patients [50]. Therefore, it is likely that the findings of our systematic review may be attributed to underreporting or death due to cancer progression before implant failure is observed in the patient population we analyzed, but further investigation is needed to identify these events and contributing factors.

In the included studies on the TS approach, there was limited emphasis on whether complications occurred more frequently after the placement of the second nail. Only Kerr et al. [26] mentioned that reported events of cardiac arrests occurred after the placement of the second nail. The rest of the studies did not specify when the complications occurred, although it should be assumed that all mortality, and specifically intraoperative mortality, occurred after the second nail. However, whether the second nail added to the existing insult from the first nail is difficult to assess, but it may also be assumed that those patients were relatively medically optimized before the second surgery.

Moreover, there was limited specification whether complications were different between impending and complete fractures. Although with a limited sample size, Ristevski et al. [1] found a lower mortality rate in patients undergoing bilateral IMN fixation for impending fractures in contrast to complete and a combination (one side complete fracture and the other side impending fracture). Similarly, but not specific for bilateral femora, in a cohort of 16 SS multiple IMNs (femur, humerus, and tibia), Moon et al. [3] reported on the low rate of mortality in the impending fracture group despite the fact that all nails were reamed and the majority of long bones were not vented.

#### 4.2.2. Same-Admission Mortality and Survivorship

Our results support the safety of a SS approach based on comparable results concerning survivorship and total same-admission mortality, including intraoperative and postoperative mortality during the same admission, to TS cases. At baseline, there is an increased risk of perioperative adverse events, particularly cardiopulmonary complications, and mortality in patients undergoing bilateral femoral IMN for metastatic bone disease compared to unilateral cases [51,52]. Historically, the fear of decreased survivorship and increased mortality rates has called against the SS bilateral IMN in favor of a TS approach for this patient population [25,26,27,28,49]. However, our results reiterate previous findings [3,7] that mortality rates for SS bilateral femoral IMN were overestimated in the past. In the available survival data for 25 patients in the SS subgroup, 14 (56.0%) patients survived 90 days or longer after the operation. By comparison, in the available survival data for 19 patients in the TS subgroup, 10 (52.6%) patients survived 90 days or longer, which was comparable (Table 3). There were no differences in the Kaplan–Meier survivorship between the two cohorts (Figure 2), similar to the previous study [7].

Moreover, in the available data on total same-admission mortality, there were no significant differences in rates in total same-admission mortality, which included a total of ten same-admission deaths, of which seven and three were in the SS and the TS cohort, respectively. These included six intraoperative deaths, of which four and two were in the SS and the TS cohort, respectively. The rate of intraoperative mortality did not differ between the SS and the TS cohort, which is a testament to the comparable safety of SS compared to TS. There were an additional four postoperative deaths during the same admission, of which three and one were in the SS and the TS cohort, respectively. The rate of postoperative mortality during the same admission was also not statistically significant [51]. There are many variables besides the surgical modality and timing, including the patient’s baseline prognosis, that affect each patient’s unique prospect of survival, and thus the definite role of surgery in mortality is difficult to ascertain.

#### 4.2.3. Blood Loss and Transfusion

No statistically significant difference was found for mean estimated blood loss between SS and TS in the one study that reported it [31]. The data on blood transfusion were not meaningful for any conclusion, but the same study showed no difference when multiple long bone IMNs are placed in a SS or TS setting, with or without bilateral femur [31]. This suggests that although SS bilateral femoral IMN is perceived as a more intensive procedure, it has no clinical implications on total blood loss and transfusion compared to TS. In our experience and an ongoing study, SS patients receive more intraoperative and postoperative transfusions, while TS patients receive more preoperative transfusions. The threshold for transfusion may be lower in anticipation of the second surgery in TS, thus accounting for a similar result.

### 4.3. Efficacy

#### 4.3.1. Length of Stay

LOS has been shown to be significantly shorter in the SS compared to the TS group (7.3 days versus 21.3 days, *p* = 0.006) [31]. Although the paper did not specify just femur, patients requiring IMNs in multiple long bones in the same admission (single-stage or multiple-stage) had significantly longer lengths of stay and took longer to initiate formal rehabilitation compared to patents requiring only one nail during one admission [31]. Moreover, overall (surgical + medical) complications, including cardiopulmonary, were higher in the multiple IMN group compared to the solitary IMN. There was no statistically significant difference in in-hospital mortality and overall survival, but the trend favored the solitary IMN group, again suggesting a lower disease burden and overall better general condition compared to the multi-IMN group.

Comparing by fracture type addressed, Ristevski et al. [1] reported a mean LOS of 39.7, 31.4, and 15.6 days for the prophylaxis, fracture, and combination subgroups undergoing SS bilateral femoral nailing procedures, respectively, although no comparison to a TS cohort was made. Similar results have also been shown in the trauma literature by Flagstad et al. [48] who compared SS vs. TS bilateral femoral IMN and noted a significantly shorter LOS for SS versus TS cases (16.4 vs. 28.5 days; *p* < 0.01). Length of stay depends substantially on attaining pain control and functional independence and the SS approach may help expedite reaching this goal due to earlier stabilization of the weight-bearing bones and resumption of weight-bearing status [3,7].

#### 4.3.2. Rehabilitation and Adjuvant Therapy

A comparison could not be made as data were not robust for analysis although one study [31] did show a shorter time to rehabilitation in SS IMN compared to multi-stage IMN for multiple bones, including but not limited to the femur. Regarding adjuvant therapy, the start of definitive adjuvant therapy for primary cancer is dependent on multiple other independent factors, and thus it may be difficult to show any difference. Nevertheless, time to rehabilitation and adjuvant therapy are important measures of efficacy of SS vs. TS, as a primary benefit of the SS approach has been stated as an expedition towards improved function and continuation of oncologic treatment for these patients.

#### 4.3.3. Functional Scores and Costs

No study has reported on functional scores or costs specific to SS vs. TS bilateral IMN in these patients. Although theoretically there should not be a difference in the final functional scores in either group, costs related to surgery may be reduced with SS due to fewer trips to the operating room, reduction of fees due to multiple procedures in the same setting, reduced complications, and reduced LOS [11,31,53]. Nevertheless, the total costs of cancer treatment extends beyond those related to orthopedic surgery, which are likely a fraction of such treatment.

### 4.4. Limitations

The current study has several limitations. First, this is a rare event. Our systematic review is limited by a relatively small and heterogeneous sample size, which is partially attributed to the uniqueness of synchronous bilateral femur impending or pathologic fractures. Additionally, several studies had listed patients who underwent SS and TS bilateral femoral IMN for treatment of oncologic indications, but these data were not clearly delineated from other indications or treatment modalities and were therefore unavailable for the purposes of our analysis [22,54,55,56,57,58,59]. Second, there is no standardized indication for staging decisions and these are primarily based on subjective factors such as a patient’s tumor and comorbidity burden, postoperative rehabilitation and adjuvant therapy, and personal, familial, or institutional preferences. Therefore, in the absence of current protocols to guide uniform staging decisions, the current study is subjected to selection bias, as sicker and/or older patients may have been skewed towards a TS approach rather than a SS approach, and thus may have higher rates of complications.

Third, there was a high rate of loss to follow-up, a small sample size, and likely differences in the definition of survival among the included studies, which prevented a robust analysis of survivorship. Likewise, variables such as rehabilitation, adjuvants, functional outcomes, and costs were rarely and not consistently collected across the literature. These variables are also pertinent contributors to oncologic outcomes, and thus our analysis may be limited in capturing the effects of other confounders to the measured outcomes of safety, efficacy, and survivorship. Fifth, outcomes were not always reported in all studies and parameters to define outcomes were heterogeneous; a standard guideline to define outcomes per procedure would strengthen the results. Sixth, patient demographics were not compared or mentioned in any study; since all these patients have advanced cancer, variables such as age, sex, BMI, ASA grade, and comorbidity index are not meaningful nor useful in establishing protocol or indications for either approach. Performance scores may be a more practical indicator and should be investigated in future studies. Finally, there was a high degree of variability in the primary diagnosis of the included patients. Although diseases such as multiple myeloma and metastatic bone disease are often surgically treated in a similar manner, they differ in disease history and prognosis, and may skew findings, although this is largely unavoidable as any randomization efforts may be unethical.

Despite these limitations, the current study offers a novel contribution to the current debate on the optimal surgical timing for patients with synchronous impending or pathologic fractures of bilateral femora secondary to oncologic indications. We do so by presenting, to the best of our knowledge, the largest and most comprehensive study of its kind in our systematic review, which provides salient evidence in the discussion to support SS bilateral femoral IMN as a relatively safe and effective surgical approach in this patient population. Future studies may use this paper as a foundation to address gaps and limitations that we have mentioned; currently, we are working on addressing this with a case series.

## 5. Conclusions

Currently, there is no consensus and/or recommendation regarding the timing of IMN for oncologic patients presenting with synchronous bilateral femoral disease. Thus, we sought to systemically review and compare the available literature on single-stage (SS) vs. two-stage (TS) bilateral femur IMN. Our systematic review supports a single-stage bilateral femoral nailing procedure as a reasonable surgical treatment strategy in select patients with synchronous complete and/or impending pathologic fractures of the bilateral femur. Compared with a two-stage approach, single-stage offered several benefits including comparable survival, same-admission mortality, and blood loss, with lower complication rates and likely shorter LOS, which supports this strategy in providing definitive early surgical fixation while aiming to expedite the adjuvant treatment and rehabilitation required in this patient population. Though ours is the most comprehensive analysis of its kind with encouraging results, larger and higher-level evidence studies are required to further delineate optimal treatment guidelines for these unique patients who are increasing in number.

## Figures and Tables

**Figure 1 cancers-15-04396-f001:**
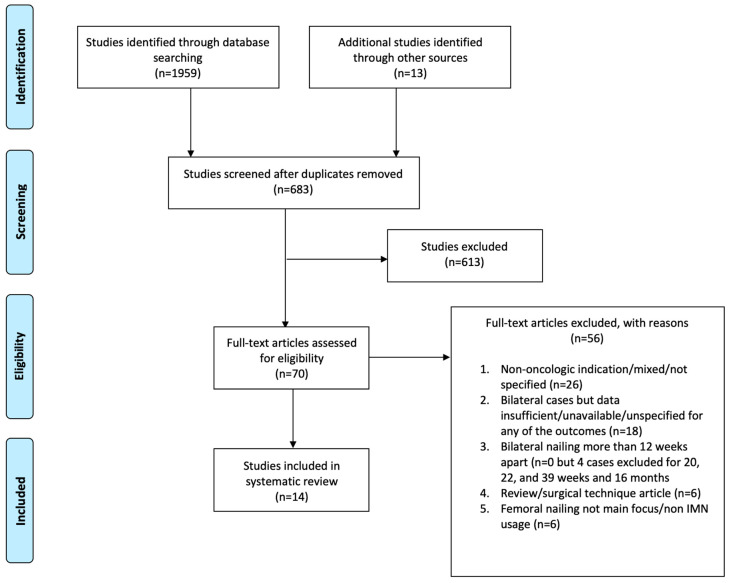
PRISMA flowchart depicting the article selection process for the investigation.

**Figure 2 cancers-15-04396-f002:**
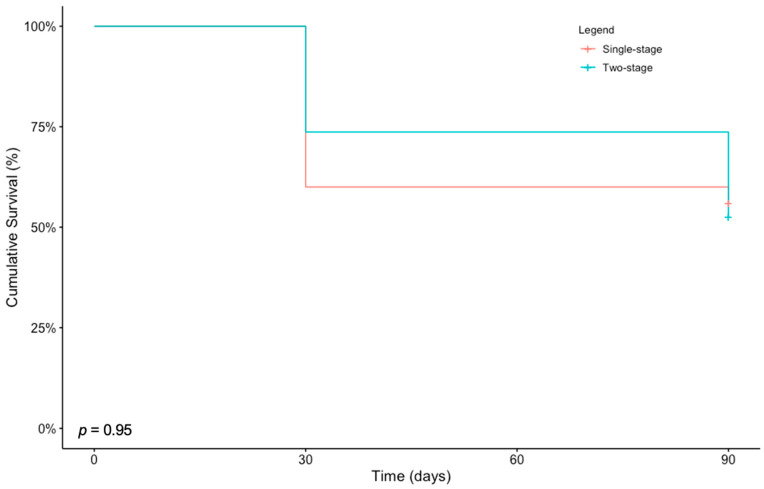
Kaplan–Meier curves depicting the survival of patients who underwent bilateral femur intramedullary nail fixation in single-stage or two-stage.

**Table 1 cancers-15-04396-t001:** General Characteristics of Included Studies.

Author (Year)	Level of Evidence	Single-Stage (SS) vs. Two-Stage (TS)	Number of Patients (n)	Gender (M:F)	Mean Age (Range)	Primary Tumor Type (n)	Fracture/Lesion Location (n)	Pathologic Fracture (n)	Impending Fracture (n)
Kerr et al., (1993) [26]	IV	SS	1	0:1	51	Breast—1	Multiple	0	1
		TS	2	0:2	64.5 (61–68)	Breast—2	Multiple	0	2
Damron et al., (1999) [33]	IV	SS	1	NS		Unknown primary—1 Prostate—1	NS	0	1
		TS	1	NS			NS	0	1
Giannoudis et al., (1999) [30]	IV	SS	3	1:2	58.67 (51–69)	Breast—2 Lung—1	Proximal—4 Middle—2 Distal—0	6	0
		TS	0						
Assal et al., (2000) [37]	IV	SS	0						
		TS	2	0:2	79 (74–84)	Lung—1 Breast—1	Multiple	NS	NS
Barwood et al., (2000) [17]	IV	SS	0						
		TS	1	0:1	NS	NS	NS	NS	NS
Cole et al., (2000) [29]	III	SS	1	NS	52	Prostate—1	NS	First nail—1 Second nail—1	0
		TS	4	NS	70.71	Breast—1 Prostate—1 Myeloma—1 Unknown—1	NS	First nail—3 Second nail—1	First nail—1 Second nail—3
Gibbons et al., (2000) [36]	III	SS	0						
		TS	7	NS	NS	NS	NS	0	7
Edwards et al., (2001) [35]	IV	SS	1	0:1	NS	Breast—1	NS	0	1
		TS	0						
Samsani et al., (2003) [21]	IV	SS	0						
		TS	3	0:3	NS	Breast—3	Subtrochanteric	NS	NS
Ristevski et al., (2009) [1]	III	SS	Total—18 Fracture—8 Prophylactic treatment—3 Combination (One side fracture, one side prophylactic): 7	Fracture—2:6 Prophylactic treatment—1:2 Combination (One side fracture, one side prophylactic): 2:5	Fracture—65.8 Prophylactic treatment—67.5 Combination (One side fracture, one side prophylactic): 60.8	Fracture: Lung—1 Breast—3 Other—4 Prophylactic: Prostate—1 Other—2 Combination (One side fracture, one side prophylactic): Lung—3 Breast—1 Other—3	Diaphyseal	15	10
		TS	0						
Moon et al., (2011) [3]	IV	SS	2 (one also had a humerus IMN in same setting)	2:0	63 (58–68)	Urothelial—1 Esophageal—1	NS	0	2
		TS	0						
Shemesh et al., (2014) [34]	IV	SS	1	0:1	67	Breast—1	Pertrochanteric—1 Subtrochanteric—1	1	1
		TS	1	0:1	64	Breast—1	Pertrochanteric/Subtrochanteric—1 Pertrochanteric—1		2
Fujita et al., (2018) [38]	IV	SS	1	0:1	64	Breast—1	Subtrochanteric—1	1	0
		TS	0						
Maheshwari et al., (2023) [31]	III	SS	7	2:5	57.6 (32–76)	MM—3 Breast—2 Granular Cell Sarcoma—1 Lung—1	Multiple	1	6
		TS	21	11:10	65.7 (42–84)	MM—10 Thyroid—1 Breast—3 Lung—3 Prostate—4	Multiple	5 (4 of which were mixed pathologic/impending)	16

NS: Not Specified.

**Table 2 cancers-15-04396-t002:** Surgical Procedure and Technique.

Author (Year)	Single-Stage (SS) vs. Two-Stage (TS)	Number of Patients (n)	Time Delay Before 2nd Procedure, n	Type of Nail (n)	Nail Size	Canal Venting (n)	Reamed vs. Unreamed (n)	Distal Locking (no. of Screws)	Reamer/Irrigator/Aspirator (RIA) Performed?	Type of Anticoagulation Used
Kerr et al., (1993) [26]	SS	1		AO nail—1	NS	No	Reamed—1	NS	NS	NS
	TS	2	9 days—1 12 days—1	Reconstruction nail—2	NS	Distal venting—1	Reamed—2	NS	NS	NS
Damron et al., (1999) [33]	SS	1		Long Gamma nail—1 All patients in this series had 130-degree implants placed	17 mm proximal diameter 11 mm distal diameter	NS	Reamed—1	Yes—2	NS	Aspirin or coumadin for 6 weeks
	TS	1	NS	Long Gamma nail—1 All patients in this series had 130-degree implants placed	17 mm proximal diameter 11 mm distal diameter	NS	Reamed—1	Yes—2	NS	Aspirin or coumadin for 6 weeks
Giannoudis et al., (1999) [30]	SS	3	NS	AO solid nail—3	9 mm nail	NS	Unreamed—3	Yes—1	NS	NS
	TS	0								
Assal et al., (2000) [37]	SS	0								
	TS	2	2 weeks—1 3 weeks—1	NS	NS	NS	Unreamed—2	NS	NS	NS
Barwood et al., (2000) [17]	SS	0								
	TS	1	3 days—1	NS	NS	NS	NS	NS	NS	Subcutaneous heparin—5000 U at time of surgery and repeated at 12 h intervals for entire hospital stay
Cole et al., (2000) [29]	SS	1		NS	NS	No	Reamed—1	NS	NS	NS
	TS	4	Average: 20.75 days Pt. 1—6 days Pt. 2—10 days Pt. 3—11 days Pt. 4—56 days	AO solid nail—2	NS	No	Reamed—2 Unreamed—2	NS	NS	NS
Gibbons et al., (2000) [36]	SS	0								
	TS	7	NS	Long Gamma nail—4 AO solid nail—3	NS	No	Reamed—4 Unreamed—3	NS	NS	NS
Edwards et al., (2001) [35]	SS	1	NS	Long Gamma nail—1	NS	NS	Reamed—1	NS	NS	NS
	TS	0								
Samsani et al., (2003) [32]	SS	0								
	TS	3	2 to 3 weeks	Long Gamma nail—3 130-degree cephalic screw	11 or 12 mm	NS	Reamed—3	NS	NS	NS
Ristevski et al., (2009) [1]	SS	Total—18 Fracture—8 Prophylactic treatment—3 Combination—7		NS	NS	NS	NS	NS	NS	NS
	TS	0								
Moon et al., (2011) [3]	SS	2	NS	NS	NS	2 *	Reamed—2	NS	NS	NS
	TS	0								
Shemesh et al., (2014) [34]	SS	1	NS	NS	NS	NS	NS	Yes	NS	NS
	TS	1	2 weeks	NS	NS	NS	NS	Yes	NS	NS
Fujita et al., (2018) [38]	SS	1	NS	NS	NS	NS	Reamed—1	NS	NS	NS
	TS	0								
Maheshwari et al., (2023) [31]	SS	7		Gamma 3 or TFNA	10	no	Diaphyseal reaming was avoided/minimized and only performed when the canal was narrow.	No	0 nails	Heparin and low-molecular-weight heparin in hospital. Aspirin × 6 weeks at Discharge if no thromboembolism
	TS	21	Average: 7.05 days (n = 21)		10	no		Yes, 1–2	4 nails	

* Moon et al. [3] described venting in two patients with multiple SS prophylactic long bone IMNs (1 femur–femur and 4 femur–humerus) but did not mention specifically for bilateral femur case. NS: Not specified.

**Table 3 cancers-15-04396-t003:** Measures of Safety and Efficacy.

Author (Year)	Single-Stage (SS) vs. Two-Stage (TS)	Number of Patients (n)	Medical and Surgical Complications, n	Infection	Reoperation	Mean Survival after Final Surgery, Weeks	Intraoperative/In-Hospital Mortality (n)	Blood Loss and Blood Transfusion	Implant Failure leading to Revision or Reoperation	Length of Stay (Days)	Time to Rehabilitation and Adjuvant Therapy (Days)
Kerr et al., (1993) [26]	SS	1	Intraoperative cardiac arrest—1 (1 fat embolism)	None	None	21.7		NS	NS	NS	NS
	TS	2	Intraoperative cardiac arrest—2 (1 air/1 fat emboli)	None	None	0	Intraoperatively—2 (Cardiac arrest) No other in-hospital deaths.	NS	NS	NS	NS
Damron et al., (1999) [33]	SS	1	NS	NS	NS	NS	NS	NS	NS	NS	NS
	TS	1	NS	NS	NS	NS	NS	NS	None	NS	NS
Giannoudis et al., (1999) [30]	SS	3	ARDS—0	NS	NS	NS	NS	NS	None	NS	NS
	TS	0									
Assal et al., (2000) [37]	SS	0									
	TS	2	None	None	None	NS	NS	NS	NS	NS	NS
Barwood et al., (2000) [17]	SS	0									
	TS	1	NS	NS	NS	NS	NS	NS	NS	NS	NS
Cole et al., (2000) [29]	SS	1	NS	NS	NS	28	NS	NS	None	NS	NS
	TS	4	NS	NS	NS	Pt. 1—5 Pt. 2—3 Pt. 3—8 Pt. 4—3	NS	NS	None	NS	NS
Gibbons et al., (2000) [36]	SS	0									
	TS	7	NS	NS	NS	NS	NS	NS	None	NS	NS
Edwards et al., (2001) [35]	SS	1	Intraoperative pulmonary embolism leading to cardiac arrest	None	None	0	Intraoperatively—1 (Cardiac arrest) No other in-hospital deaths.	NS	NS	NS	NS
	TS	0									
Samsani et al., (2003) [32]	SS	0									
	TS	3	NS	NS	NS	NS	NS	NS	NS	NS	NS
Ristevski et al., (2009) [1]	SS	Total—18 Fracture—8 Prophylactic treatment—3 Combination—7 (fracture one one side and impending on other side)	DVT within 3 months: Fracture—0 Prophylactic—0 Combination—0	None	None	Total died by 3 months: 7 pts (4 fracture, 0 prophylactic, 3 combination)	Total In-hospital death: 5 pts (3 fracture, 0 prophylactic, 2 combination group) Intraoperative Death: 3 pts (2 fracture, 0 prophylactic, 1 combination) Postoperative, Same-admission: 2 pts (1 fracture, 0 prophylactic, 1 combination) No reasons were reported for in-hospital mortality.	NS	NS	Fracture—31.4 Prophylactic—39.7 Combination—15.6	NS
	TS	0									
Moon et al., (2011) [3]	SS	2	Postoperative respiratory distress from presumed fat emboli—1 *	None	None	1 Pt.—2 1 Pt.—>4	Postoperative, same-admission—1 (2 weeks)	NS	NS	NS	NS
	TS	0									
Shemesh et al., (2014) [34]	SS	1	NS	None	None	78.2	NS	NS	None	7.3 ± 4.5 (range, 1–14)	NS
	TS	1	Infection that required debridement	Deep wound infection	Surgical Debridement	47.8	NS	NS	None	21.3 ± 18.1 (range, 3–65)	NS
Fujita et al., (2018) [38]	SS	1	Intraoperative cardiac arrest due to primary tumor/fat embolism with successful resuscitation	None	None	NS	NS	Blood loss: 550 ml	NS	NS	NS
	TS	0									
Maheshwari et al., (2023) [31]	SS	7	Hypotension—1	None	None	Excluding those lost to follow-up (n = 1): 94.6	No intraoperative or postoperative, same-admission deaths.	Blood Loss: average: 531 ± 198 (range, 400–1000)	None	7.3 ± 4.5 [range, 1–14] days	Rehabilitation: 1 Pt.—3 days Adjuvant Therapy: 1 Pt.—14 days
	TS	21	Bilateral pleural effusion—1 Hypotension—1 Respiratory distress—1 PE, multi organ failure due to POD—1 Pneumonia—1 Urinary tract obstruction and UTI—1 AKI—1 UTI—1	None	None	Average: 55.1 (3–162.6) (n = 10) Excluding those lost to follow-up (n = 12): 30 days—8.3 90 days—25.0 365—50.0	No intraoperative deaths. 1 postoperative, same-admission death at 22 days postop.	Blood Loss: average: 419 ±226 (range, 100–1000)	None	21.3 ± 18.1 [range, 3–65] days	Rehabilitation: 16 Pt.—3.2 ± 2.0 (range, 1–8) Adjuvant Therapy: 15 Pt.—27.5 ± 12.5 (range, 9–55)

* Although not specifically for bilateral femur (but for a group of SS multiple long bone IMNs) this paper reported on 4 other non-pulmonary postoperative complications (ileus, transient chest pain, myocardial infarction, and atrial fibrillation). All recovered and were discharged after appropriate medical management. NS: Not Specified. There were no surgical complications in the SS but 1 (4.2%) deep wound infection requiring return to the operating room for surgical debridement occurred in the TS group (*p* = 0.86) [34] (Table 3).

## Data Availability

The data presented in this study are available on request from the corresponding author. The data are not publicly available due to privacy and ethical considerations.

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
