# Peer review of "Safety and Efficacy of a Single-Stage versus Two-Stage Intramedullary Nailing for Synchronous Impending or Pathologic Fractures of Bilateral Femur for Oncologic Indications: A Systematic Review"

_cancers, 2023, doi:10.3390/cancers15174396_

Round 1

Reviewer 1 Report

Bravo for looking into this topic - definitely a consideration for this unique subset of patients in orthopaedic oncology.  I do think this manuscript has potential for acceptance; however, I do have several areas that I would like to see addressed before I would support the acceptance for publication.

1. Time range - I think that this goes too far back in time.  Including studies from 1993 is far too long ago - you said it yourself, there have been so many advances in oncology from a medical (systemic therapy) standpoint and from a surgical standpoint, it is not appropriate to go back this far.  So then, how far is too far?  I don't have the final answer, but I would suggest 10 years as a reasonable maximum retrospective look. 

2. Tables are very difficult to read and compare as they are presented in this manuscript.  Turn them into landscape, if possible, or completely redesign them so they are easier to read and compare.  Until these are fixed, I would absolutely reject this article.

3. Please further emphasize the high rates of complication with synchronous bilateral disease.  You show intraop + same admission mortality that is very high.  Please give some comparison of the synchronous bilateral disease compared to what is reported for single site disease that is well documented in the literature. 

4. Too much time is spent detailing/explaining all the variables you looked at that had insufficient data to assess, such as outcomes measures, blood less, etc.  Just say you looked at those and you couldn't compare.  Showing up in methods, data and discussion is a waste of a lot of words/characters in the manuscript.

5. Re: the type of nail - some of the nails used in this aren't even on the market any longer, so using the specific type is not really beneficial.  Solid vs cannulated.  Reamed vs unreamed.  Those are reasonable, but mentioning the Stryker Gamma by name has no meaning here.  You don't even really discuss "hip fracture" nail vs diaphyseal nail, both of which can be used depending on lesion or fracture location or patient anatomy are more meaningful, but probably still not relevant to this manuscript.   

6. You are correct to highlight their is no way to really compare the patients who got SS vs TS.  I am on the fence about the significance of this.  If the groups are systematically different (there are significant differences between them), then there is basically no conclusion that can be drawn from this manuscript, meaning any attempt to compare the timing of IMN is potentially futile.  I do wonder if this should just be a systematic review of bilateral femoral nail and not a comparison at all.  

Reviewer 2 Report

The scientific paper "Safety and Efficacy of a Single-Stage versus Two-Stage Intramedullary Nailing for Synchronous Impending or Pathologic Fractures of Bilateral Femur for Oncologic Indications: A Systematic Review of Literature" aimed to conduct an updated systematic review of existing literature to analyze and compare the safety and efficacy of a SS approach to a TS approach for patients presenting with synchronous bilateral impending and/or pathologic fracture of the femur for oncologic indications.

It can be considered that:

1)      Remove abbreviations next to the authors' names in the title of the manuscript.

2)      Remove "of literature" from the title of the manuscript.

3)      Adjust handwriting text alignment to "justified"

4)      Insert tabs at the beginning of paragraphs

5)      Separate the paragraph between lines 43 and 67 into 2 or 3 paragraphs.

6)      Include in the methodology how to avoid the risk of bias in the selection of analyzed manuscripts?

7)      What version of R Statistical Software is used? Insert manufacturer, city, state and country

8)      In the Prisma chart, figure 1, in excluded articles (n=56), add n=0 in item 3

9)      In the Prisma chart, figure 1, in the articles included for analysis (n=14) two new data SS n=72 and TS n=84 appear. How were these numbers obtained? It was strange because the n informed was 14. Wouldn't it be recommended to remove this and leave only the explanation in the text that is immediately after in the manuscript?

10)  I recommend placing the tables with the largest pages, in landscape format, to facilitate understanding. In its current form, it's hard to read.

11)  In the tables, add in the author and year column, the number of the reference

12)  Start the conclusions by adding a general context of the purpose of the systematic review carried out

13)  Add more references to the discussion.

Minor editing

Reviewer 3 Report

In this systematic review, Nian et al. revised several works reporting the treatment of femur fractures occurring in oncologic patients. The review is written in good english and each section is presented clearly. The main findings are described in detail and nicely organized per topic. 

The paper provides a noteworthy contribution to the scientific literature, although the study limitations correctly reported by the Authors in a proper paragraph.

I only suggest to add a graphical abstract to attract readers and summarize with pictures the manuscript content.

Author Response

Point 1: In this systematic review, Nian et al. revised several works reporting the treatment of femur fractures occurring in oncologic patients. The review is written in good english and each section is presented clearly. The main findings are described in detail and nicely organized per topic.

The paper provides a noteworthy contribution to the scientific literature, although the study limitations correctly reported by the Authors in a proper paragraph.

I only suggest adding a graphical abstract to attract readers and summarize with pictures the manuscript content. 

Response 1: A graphical abstract has been added. Please see the attachment. Also, the PRISMA flowchart has been made simple and clear to follow.

Round 2

Reviewer 2 Report

-/-

 Minor editing of English language required